# The Molecular Resistance Mechanisms of European Earwigs from Apple Orchards Subjected to Different Management Strategies

**DOI:** 10.3390/insects14120944

**Published:** 2023-12-13

**Authors:** Thierry Fricaux, Adrien Le Navenant, Myriam Siegwart, Magali Rault, Christine Coustau, Gaëlle Le Goff

**Affiliations:** 1Université Côte d’Azur, INRAE, CNRS, ISA, F-06903 Sophia Antipolis, France; thierry.fricaux@inrae.fr (T.F.); christine.coustau@inrae.fr (C.C.); 2Avignon Université, Aix-Marseille Université, CNRS, IRD, IMBE, Pôle Agrosciences, 301 rue Baruch de Spinoza, BP 21239, F-84916 Avignon, France; adrien.le-navenant@inrae.fr (A.L.N.); magali.rault@univ-avignon.fr (M.R.); 3INRAE, Unité PSH, Site Agroparc, F-84914 Avignon, Cedex 9, France; myriam.siegwart@inrae.fr

**Keywords:** *Forficula Auricularia*, orchard managements, insecticides, acetylcholinesterase, nicotinic acetylcholine receptors, detoxification genes

## Abstract

**Simple Summary:**

Apples are among the most heavily treated fruits against pests. Finding new, more environmentally friendly solutions for pest control is essential. Biological control is one of these, and can involve the use of predatory insects to eliminate pests. The earwig is one such predatory insect. We set out to determine whether earwigs had developed resistance mechanisms to the molecules to which they are exposed in orchards. A single population of earwigs was sampled in three types of orchard (organic, conventional and integrated pest management) and compared with each other. Mutations and higher expression levels in genes known to confer resistance were identified. These results highlight the effect of pesticide use on beneficial organisms and the resulting effect on the biodiversity of natural enemies in orchards.

**Abstract:**

To date, apple orchards are among the most treated crops in Europe with up to 35 chemical treatments per year. Combining control methods that reduce the number of pesticide treatments is essential for agriculture and more respectful of the environment, and the use of predatory insects such as earwigs may be valuable to achieve this goal. European earwigs, *Forficula auricularia* (Dermaptera: Forficulidae) are considered beneficial insects in apple orchards where they can feed on many pests like aphids. The aim of this study was to investigate the potential impact of orchards’ insecticide treatments on resistance-associated molecular processes in natural populations of earwigs. Because very few molecular data are presently available on earwigs, our first goal was to identify earwig resistance-associated genes and potential mutations. Using earwigs from organic, integrated pest management or conventional orchards, we identified mutations in acetylcholinesterase 2, α1 and β2 nicotinic acetylcholine receptors. In addition, the expression level of these targets and of some essential detoxification genes were monitored using RT-qPCR. Unexpectedly, earwigs collected in organic orchards showed the highest expression for acetylcholinesterase 2. Four cytochromes P450, one esterase and one glutathione S-transferases were over-expressed in earwigs exposed to various management strategies in orchards. This first study on resistance-associated genes in *Forficula auricularia* paves the way for future experimental studies aimed at better understanding the potential competition between natural enemies in apple orchards in order to optimize the efficiency of biocontrol.

## 1. Introduction

As long-term perennial crops, unsprayed apple orchards provide stable ecological habitats and can host a great biodiversity of up to 2000 arthropod species [1,2]. This biodiversity is strongly impacted by pesticides, reducing both pests and their natural enemies, especially in apple orchards which are among the most heavily treated crops [3]. Rethinking control strategies involves reducing the use of pesticides to promote the presence of natural enemies and their contribution to biological control in fruit crops. Moreover, some authors suggested the use of pesticide-resistant natural enemies in combination with a reduced application of chemicals as an interesting alternative. A successful example has been reported for the control of two apple pests, *Panonychus ulmi* and *Aculus schlechtendali*, with the use of organophosphate-resistant strains of the predatory phytoseiid mite *Typhlodromus pyri* Scheuten [4]. A recent study showed that field-collected predatory mites *Neoseiulus californicus* McGregor were 27-fold more resistant to pirimicarb than a susceptible reference population, and this resistance factor reached 69-fold after five generations of laboratory selection. Most interestingly, the authors showed an increased ability of the selected F5 population to consume adults of *Tetranychus urticae* as compared to the parental population [5]. This could lead to improved integrated mite management programs by using pesticide-resistant *N. californicus* in apple orchards.

The resistance mechanisms that develop following repeated exposure to insecticides are similar for pests and their natural enemies, although this is less studied for in the latter. The two most common mechanisms for acquiring resistance include a modification of the insecticide target or the development of a metabolic resistance involving the degradation of the insecticide via detoxification enzymes [6]. Apple orchards are treated with nearly all the chemical families available on the market. The most commonly used are organophosphates (OPs), neonicotinoids and synthetic pyrethroids [7,8] which all act on the insect nervous system. Resistance-associated mutations have been reported for the targets of each of these pesticide families. OPs and carbamates inhibit acetylcholinesterase (AChE) and several mutations in one or the two genes coding for AChE have been demonstrated to play a role in resistance (reviewed in [9]). Neonicotinoids act on the nicotinic acetylcholine receptors (nAChR) and resistance has often been related to a disruption in the different subunits of these receptors [10,11]. An example from the aphid *Myzus persicae* is the mutation R81T in nAChR β1 subunit that reduces the neonicotinoid binding [12]. The voltage-sensitive sodium channel is the target of pyrethroids and the two more frequent mutations identified to account for resistance are kdr at L1014 and super-kdr at M918 [13]. Many other mutations have also been reported. For example, in their review, Feyereisen et al. [9].mentioned 61 different mutation positions in 51 different species.

In addition to the mutated pesticide target, the second major process for resistance is the involvement of detoxification enzymes. Several types of enzymes modify the generally lipophilic insecticide molecules into a more hydrophilic molecule that is easily excreted from the body. This process can be sequential and involves functionalisation enzymes like cytochromes P450 (P450s) and carboxyl/choline esterases (CCEs). The resulting metabolites can then be excreted or taken over by so-called conjugation enzymes such as glutathione S-transferases (GSTs) and UDP-glycosyl transferases (UGTs). The final step involves efflux transporters such as ATP-binding cassette transporters (ABC) to pump the molecule out of the cell. There are many examples of resistance to major pesticide families involving detoxification enzymes. The most common mechanisms correspond to the over-expression of these enzymes, which allow them to metabolize a greater amount of insecticides. For example, in an orchard pest such as *Cydia pomonella*, it has been shown that a P450, CYP6B2, is over-expressed in OP and pyrethroid-resistant strains, with overexpression levels in Raz and Rv reaching 241.4- and 77.3-fold, respectively [14]. When CYP6B2 expression is reduced via RNAi, sensitivity to the insecticides, azinphos-methyl and deltamethrin is increased [14]. P450s are also involved in neonicotinoids resistance. An emblematic example is found in the aphid *Myzus persicae nicotianae* where resistance to neonicotinoid is due to the overexpression of a single P450, CYP6CY3 [15]. The origin of this overexpression has been identified and corresponds to gene amplification and the expansion of a dinucleotide microsatellite in the promoter region of CYP6CY3. Furthermore, the authors demonstrated that CYP6CY3 metabolizes neonicotinoids to less-toxic compounds [15].

The European earwig *Forficula auricularia* is a major and voracious natural enemy of insect pests in apple orchards. As a generalist predator, it has been shown to feed on insects belonging to several orders such as Coleoptera, Collembola, Diptera, Lepidoptera, Hemiptera and Hymenoptera [16]. For example, it can be efficient in suppressing the woolly apple aphid, *Eriosoma lanigerum* (Hausman) [4]. Experiments showed that earwig larvae could consume up to 1000 psyllid eggs per day [17,18], and up to 8700 green apple aphids (*Aphis pomi* Degeer) in the course of the full earwig larval development [19]. It has been shown that in orchards with a long history of chemical insecticides application, earwigs have developed resistance [20,21]. For example, we recently reported that earwigs collected in orchards treated with conventional methods were more resistant to chlorpyrifos-ethyl (OP insecticide) than those collected in organic or integrated pest management orchards [20]. More specifically, constitutive GST and CCE activity levels were higher, and a decrease in sensitivity of AChE was observed in male and female earwigs pre-exposed in the conventional orchard [20]. However, the molecular mechanisms behind these observed variations in enzymatic activity or reduced sensitivity of acetylcholinesterase have not been investigated until now and were the aim of the present study.

Based on the diversity of pesticides applied to orchards, we hypothesized that repeated and diverse treatments may result in the selection of various pesticide resistance traits in earwigs. The purpose was to better understand how such selection could affect natural enemies in orchards. The two main resistance mechanisms were a change in the insecticide target or increased metabolism of the insecticide. We investigated these two types of mechanisms at the molecular level, first by looking for the presence of mutations in the targets of insecticides used in orchards, and second by monitoring the level of expression of these same targets as well as selected detoxification enzymes known to be involved in resistance. 

## 2. Materials and Methods

### 2.1. Insect Sampling

Insects used in the present study came from the same sampling described in Le Navenant et al. [20]. Briefly, adult earwigs from both sexes were collected in apple orchards in Noves close to Avignon (South-eastern France) in July 2017. They were trapped using cardboard placed around the trunk of the apple trees in an INRAE experimental orchard without any pesticide treatment (no treatment, NT), and in three commercial orchards either under chemical treatments (conventional, CONV) or under integrated pest management (IPM), where pesticides were applied according to damage thresholds, and, finally, in an organic (ORG) orchard where chemical pesticides had been replaced by organic products. In each commercial orchard (ORG, IPM and CONV) treatment calendars provided by the farmers were recorded for the last ten years. 

### 2.2. The Cloning of Genes of Interest for Studying Insecticide Resistance

In order to study the molecular mechanisms of resistance, the sequences of genes coding for the targets of insecticides and some detoxification enzymes, as well as 3 genes used as reference in the RT-qPCR experiments, were searched for in insect species phylogenetically related to the earwig. Indeed, the genome of *F. auricularia* was not available when we started our study, but it has been sequenced since [22,23]. Earwigs belong to the order Dermaptera and the superorder Polyneoptera [24]. In the same superorder, crickets and cockroaches (belonging to Orthoptera and Blattodea, respectively [24]), offered the advantage of having gene sequences available. These sequences allowed us to perform a BLAST search on transcriptome data of *F. auricularia* [25]. Primers were designed from consensus EST sequences and were used to amplify and clone these genes of interest from the collected earwigs. Genes were cloned and sequenced several times. Validated sequences were submitted to GenBank (accession numbers are indicated in Table 1). The conditions of PCR amplification and primer sequences have been reported in Appendix A. PCR products were purified from agarose gel using GenElute Gel Extraction kit (Sigma-Aldrich, St. Louis, MO, USA), and then sub-cloned in pGEM-T easy vector systems (Promega Corporation, Madison, WI, USA). JM109 competent cells (Promega Corporation, Madison, WI, USA) were transfected with each construction of interest. Bacterial clones were purified using GenElute Plasmid Miniprep Kit (Sigma-Aldrich, St. Louis, MO, USA) and sequenced by GATC.

### 2.3. The Detection of Mutations in the Insecticide Targets

In order to search for the presence of mutations in the insecticide targets, a pool of RNA from earwigs collected in the different orchards was used. Total RNA was extracted from 4 females and 4 males individually in each orchard (NT, ORG, IPM or CONV) using TRI-reagent (Sigma-Aldrich, St Louis, MO, USA) according to the manufacturer’s instructions. Tissue homogenisation was performed in 1 mL TRI-reagent with three stainless beads of size 3.2, 4 and 8 mm (Marteau & Lemarié, Sorbiers, France) using vibrating mill MM400 (Retsch, Eragny sur Oise, France). cDNA was synthesized from 5000 ng of RNA corresponding to 156.25 ng from each earwig (8 insects by orchard, 4 management strategies) using iScript cDNA Synthesis kit (Biorad, Hercules, CA, USA). PCR amplification conditions and cloning were performed as previously described. Ten clones of each gene were sequenced by GATC. The sequencing was performed in both directions with primers defined every 500 to 600 base pairs.

### 2.4. Quantitative Real-Time PCR

Resistance may also be caused by variations in the expression levels of targets or detoxification enzymes involved in the metabolism of insecticides. Therefore, RT-qPCR experiments were performed to assess expression of those genes. Total RNA was extracted from each adult earwig using TRI-reagent (Sigma-Aldrich, St Louis, MO, USA) according to the manufacturer instructions. For each type of orchard, three RNA mixes were prepared from 80 ng of RNA from 5 females and 5 males. Each mix corresponding to 800 ng of total RNA was reverse-transcribed using iScript cDNA Synthesis kit (Biorad, Hercules, CA, USA). qRT-PCR reactions were carried out on an AriaMx Real Time PCR system (Agilent, Santa Clara, CA, USA) using Takyon No Rox SYBR MasterMix blue dTTP (Eurogentec, Seraing, Belgium). The PCR conditions were as follows: 95 °C for 3 min, followed by 40 cycles of 95 °C for 10 s, 58 °C for 20 s and 72 °C for 20 s. Each reaction was performed in triplicate and the mean of three independent biological replicates was calculated. The results were normalized using three reference genes, actin, elongation factor 1 (EF1) and glyceraldehyde-3-phosphate dehydrogenase (GAPDH) and took into account the efficiency of each primer pairs (see Appendix A for primer details). Primers were designed using primer 3 (https://primer3.ut.ee). The relative expression values were calculated using the SatqPCR tool (http://satqpcr.sophia.inra.fr/cgi/tool.cgi, accessed on 4 December 2023) and statistical differences using one-way ANOVA followed by Tukey test.

## 3. Results and Discussion

### 3.1. Comparing Treatments between Orchards

The treatments carried out in the three types of orchards (ORG, IPM and CONV) are reported in Table 2. Earwigs were also collected from a fourth orchard (NT), which served as a control, where no treatments were carried out. ORG and IPM orchards receive about 12 treatments per year while in CONV orchard this number reaches up to 20 treatments per year. The molecules used are also different according to management strategies. Granuloviruses and spinosyns are the two most commonly applied products in ORG orchards, whereas treatments in IPM orchards are based on a larger number of chemical families including pyrethroids, organophosphates, neonicotinoids and mineral oil. In CONV orchards, organophosphates were the main chemical treatment until 2016, followed by neonicotinoids. Some molecules belonging to different insecticide families can act on the same target in the insect. For example, the neonicotinoids applied either in conventional or in integrated fruit production target the nicotinic acetylcholine receptors. The spinosad used in organic orchards is a mixture of the two toxins spynosin A and spynosin D, produced by fermentation via a soil bacterium Saccharopolyspora spinose. Spinosad toxins also target nicotinic acetylcholine receptors.

### 3.2. The Modification of Insecticide Targets

Knowing the insecticides most commonly used in the orchards, the presence of mutations in the sequence of genes coding for these insecticide targets have been searched.

#### 3.2.1. Acetylcholinesterases, the Target of Carbamates and Organophosphates

Carbamates and organophosphate insecticides act on the insect nervous system and affect acetylcholinesterase. This enzyme ensures the hydrolysis of acetylcholine into choline and acetic acid at the cholinergic synapses. Several mutations in this essential protein have been identified in insects and some of them were demonstrated to be responsible for resistance (for review [9]). In F. auricularia, two genes code for acetylcholineserase *ace*1 and *ace*2 whereas high Diptera-like Drosophila melanogaster has only one gene [26]. Our sequencing study did not detect any mutation in *ace*1 but allowed us to detect one mutation in *ace*2 (Table 3). This mutation corresponds to a change of the glutamine at the position 337 toa lysine (Q337K), (corresponding to the position Q393 of Torpedo californica AChE) and has never been reported to date in other insects. However, according to the position in T. californica, this mutation is very close to the G396S mutation identified in the pest olive fruit fly, Bactrocera olea, and in the closely related species Bactrocera dorsalis that alters the structure of the enzyme and confers resistance to OPs [27,28]. G396S is close to one of the three amino acids that form the catalytic triad. Vontas et al. [27].suggested that any amino acid change in this region should produce a structural change. It is therefore likely that Fa Q337K described here could explain the decrease in sensitivity of AChE to chlorpyrifos observed by Le Navenant et al. (2019). Acetylcholinesterase is known to carry many mutations that can confer resistance, each making a small change to affect insecticide binding while maintaining the ability to hydrolyse acetylcholine.

#### 3.2.2. Nicotinic Acetylcholine Receptors, the Target of Neonicotinoids and Spinosad

Nicotinic acetylcholine receptors are pentameric transmembrane proteins that belong to the family of ligand-gated ion channels and act at cholinergic synapses. Each subunit has four transmembrane regions with a large N-terminal extracellular domain that contains loops A to F, crucial for the binding of acetylcholine and agonists [29]. A receptor can be homo- or heteromeric, composed of several subunits. For example, in the model insect *D. melanogaster*, there are seven α subunits (α1 to α7) and three β subunits (β1 to β3) [30]. The number of subunits is unknown in *F. auricularia,* but most insects display a similar number as *Drosophila* [31]. In insects, mutations in several different subunits have been reported to confer resistance to neonicotinoids [32], but only mutations in the α6 subunit have been reported to confer resistance to spinosyns [33]. However, we were able to clone two nAChR subunits, the α1 and β2. We observed two independent mutations in the α1 subunit: one involved a change from proline to a histidine at the position 145 (α1 P145H) (Table 3, Appendix A) and the other changed glutamic acid to lysine at the position 546 (α1 E546K). The α1 P145H mutation is located in the N-terminal extracellular region, which contains the ligand-binding domain (either acetylcholine or neonicotinoids) and the proline is the first amino acid after loop E [34]. Several residues of loop E have been shown to interact directly with nicotine and neonicotinoids [35,36,37]. The proximity of α1 P145H to the loop could produce a structural rearrangement and disrupt insecticide binding. To our knowledge, this mutation was not reported until now. However, because this amino acid is very well conserved from insect to human [36] (Appendix A), it suggests that it is important in the receptor function and may have an effect on resistance. The second mutation is located at the C-terminal end of the protein, which was not identified as an important region for the binding of neonicotinoids nor spinosyns. However, its role in resistance cannot be excluded without complementary analyses. The subunit β2 was also sequenced and two independent mutations have been identified (Table 3). In one clone, one mutation corresponded to the change of a serine to a proline at the position 6 (β2 S6P). This position is not very well conserved but its location in the N-terminal domain may affect insecticide binding. The second mutation was found at the position 102 and corresponded to the change of a glycine to arginine (β2 G102R). This amino acid is well conserved in insects and in proximity to loop A where a tyrosine has been shown to interact directly with imidacloprid (Appendix A) [36]. The mutations identified in the α1 and β2 subunits did not correspond to the residues shown to interact directly with neonicotinoids but were, however, close to loops A and E. This could potentially influence the structure of the protein and thus the interactions with insecticides. Further experiments are required and the recent achievement of the expression in xenopus oocytes of functional insect nicotinic acetylcholine receptors opens the way to a detailed understanding of the role of these mutations [38]. 

#### 3.2.3. The Voltage-Sensitive Sodium Channel, the Target of Pyrethroids

The gene coding for *para*, the voltage-sensitive sodium channel, was cloned in three fragments. Despite our efforts, we were unable to obtain the full coding sequence, probably due to the high complexity of the gene. It codes for a protein with four repeated domains (I to IV), each having six transmembrane segments. In *Drosophila*, the locus *para* was shown to be large and complex with at least 26 exons and alternative splicing, increasing the diversity of the produced transcripts [39]. Resistance to pyrethroids was first identified in the housefly with the presence of two-point mutations detected and named *kdr* for knockdown resistance and *super-kdr* [40]. Those mutations correspond to the replacement of a leucine at the position 1014 to a phenylalanine (L1014F) and change of a methionine at the position 918 to a threonine (M918T), respectively. Additional mutations were reported to confer resistance, mainly present in domains II and III (see for review [13]). In our study, the fragment 2 contained this domain II, but no mutation was observed. Mutations were found in fragment 3, but at poorly conserved positions and probably corresponding to genetic variability rather than resistance. One mutation was identified in domain III, the transmembrane segment 3, corresponding to the replacement of an alanine to a valine (A1375V). And, three additional mutations were found in the C-term region: arginine to proline R1879P, isoleucine to methionine I1903M and glutamic acid to lysine E1976K (Table 3). None of these positions have been identified as conferring resistance to date. In the fragment 1 (N-term and domain I), we did not observe any point mutation either. However, sequence variation was detected, in particular an insertion of 13 nucleotides between positions 148–149 (corresponding to the intracellular N-terminal loop), which is probably linked to alternative splicing.

#### 3.2.4. Other Targets

No mutation was detected in the glutamate-gated chloride channel (GluCl), the target of avermectins. Similarly, we were unable to determine the presence of mutations in the gene coding of the ryanodine receptor, the target of anthranilic diamides. This is likely because it is a very large gene (more than 15,000 nucleotides in most insects) and we were not successful in obtaining the full sequence.

### 3.3. The Modification of Expression Levels Associated with Resistance

In addition to point mutations observed in insecticide targets, resistance may be caused by variations in the expression levels of insecticide targets or detoxifying enzymes involved in insecticide metabolism. Therefore, RT-qPCR experiments were performed to assess gene expression in insecticide targets (Figure 1) as well as detoxification enzymes (Figure 2).

#### 3.3.1. The Expression of Insecticide Targets

Ace1 expression is stable between conditions NT, IPM and CONV while its expression is two times higher in organic samples (Figure 1). A duplication in ace1 has been identified in several arthropods from mosquitoes to the spotted spider mite [41,42,43]. It was suggested that the duplication could compensate the fitness cost associated with the Ace point mutation [43,44]. The variations in expression between orchards are even more marked for ace2. A two-fold factor was observed for earwigs from IPM and CONV while a factor close to two was detected in earwigs from organic orchards. Point mutations were only identified in ace2 in the present study (Table 3); the compensation mechanism may explain the level of expression observed. These results are related to those obtained in some *Drosophila* strains where quantitative and qualitative changes in ace2 are observed in the resistance to OPs and carbamates [45,46].

The expression of the three nAChR subunits α1, α3 and β2 is induced by at least by three-fold in earwigs coming from treated orchards compared to NT, with the highest induction being seen in ORG samples. The data available to date remain controversial, as several studies reported a decrease in the expression of some nicotinic acetylcholine receptor subunits associated with resistance whereas others showed an increase. Because most of these studies examined a limited number of subunits, a possible explanation of this discrepancy could be compensatory changes in the expression of different subunits [47].

For example, a reduction in the expression of a subunit involved in the binding of the insecticides could be compensated by the over-expression of other subunits. nAChRs function as pentamers of either α subunits or a combination of α and β subunits. The subunit composition may vary between susceptible and resistant insects. Our results are similar to those obtained in the small brown planthopper, *Laodelphax striatellus*, where insects resistant to the neonicotinoids imidacloprid or cycloxaprid showed higher expressions of α1 and β1 [48,49]. RNAi, targeting the α1 subunit, decreased the resistance to cycloxaprid [48]. Spinosad, used in organic orchards, targets nAChRs. Resistance has been associated with the α6 subunit and often to the introduction of a premature stop codon leading to a truncated subunit [50,51,52]. Our results show the highest expressions of α1, α3 and β2 in earwigs collected in organic orchards. We do not know if earwigs possess resistance to spinosad and if this high expression is a compensatory consequence of the down-regulation of the α6 subunit. Unfortunately, we were unable to clone the α6 subunit of *F. auricularia*. However, a general observation, rather unexpectedly, was that earwigs from organic orchards show an over-expression of insecticide targets, but this needs to be confirmed by sampling in a large number of orchards. This could result from the exposure of earwigs to spinosad in organic orchards as a compensatory response. It is known that spinosad could decrease the predation activity of earwigs [53] and cause severe neurodegeneration in *Drosophila* [54]. As a consequence, we cannot exclude a receptor up-regulation that could depend on spinosad concentration in the nervous system as it has been shown in the presence of high nicotine concentration [55]. 

#### 3.3.2. The Expression of Detoxification Enzymes

A first general observation is that the expression levels of resistance-related transcripts are higher in earwigs from treated orchards (except for CCE clade B) when compared with non-treated earwigs (Figure 2). This expected result confirms that both the orchard sampling and the detoxification genes identified and studied here are appropriate even if a larger number of orchards representative of each management condition would be desirable. The highest expression of CYP4G189 (2.57-fold) was observed for earwigs collected from CONV orchards, and the expression was slightly lower for IPM and even lower for ORG (Figure 2). CYP4G189 belongs to the CYP4G sub-family which has been identified as the enzyme responsible for the final step in the synthesis of cuticular hydrocarbons [56,57], which play a major role in the protection against desiccation and chemical communication. 

The involvement of P450s of the 4G sub-family in cuticle modification has been shown in insecticide resistance. The reduced penetration of insecticides is thought to contribute to the resistance phenomenon [58]. For example, in the mosquito *Anopheles gambiae,* the overexpression of *CYP4G16* and *CYP4G17* in pyrethroid-resistant strains was associated with an increase in cuticular hydrocarbon content and significantly thicker cuticle layers when compared to susceptible strains [59,60]. In both *Locusta migratoria* and *Nilaparvata lugens,* silencing the two genes coding for CYP4G, *LmCYP4G62* and *LmCYP4G102*, or *NlCYP4G76* and *NlCYP4G115*, increased penetration and susceptibility to insecticides [61,62]. It would be interesting to compare the cuticle thickness of the earwigs according to their orchard of origin by microscopic analysis. However, P450s are best known for their direct role in insecticide metabolism. Among them, the CYP6 family has been identified in many cases in mosquitoes, drosophila, aphids, lepidopterans and many others (reviewed in [63]). In the present study, three CYP6s have been cloned, CYP6NP1, CYP6NQ1 and CYP6NW1. The highest expression of CYP6NP1 and CYP6NQ1 (around 5-fold) was observed in the earwigs from CONV orchards whereas CYP6NW1 expression was higher in earwigs from IPM and ORG (Figure 2). Although many studies have shown a correlation between CYP6 overexpression and resistance, few studies have functionally validated the ability of these enzymes to detoxify an insecticide, as shown by the analysis of the available data in the genus *Spodoptera* [64]. Few examples also demonstrate the role of CYP6 in the resistance to OPs: in *Helicoverpa zea*, diazinon is metabolized by CYP6B8 [65] while it is the CYP6G3 that does so in *Lucilia cuprina* [66]. The role of CYP6NW1 should also be investigated, as it is the most over-expressed in earwigs from ORG and IPM, although there is almost no treatment in common between these two types of orchards (Table 2). IPM used a combination of chemical treatments, mainly neonicotinoids, pyrethroids and Ops, whereas, in ORG, it was principally granulovirus and spinosad. While CYP6s are well known to be involved in the resistance to the pesticides used in IPM, little is known about their role in resistance to the products used in ORG. The involvement of P450s has been suggested for spinosad resistance in field strains of the cotton bollworm *Helicoverpa armigera* collected in China. The use of piperonyl butoxide, a P450 inhibitor, synergized the toxicity of spinosad and P450 activities were increased after a spinosad treatment [67]. In *Musca domestica*, CYP6G4 was constitutively highly expressed in spinosad-resistant compared to susceptible strains and this P450 was induced in the susceptible strain when flies were exposed to the insecticide [68]. The role of these three induced CYP6s in *F. auricularia* needs to be further investigated.

We were able to identify two genes coding for carboxylesterases, one from clade B the other from clade E. Insect carboxylesterases are classified in three main phylogenetic classes: dietary/detoxification which contains clade A to C, hormone/semiochemical processing from clade D to G and neuro/developmental functions from clade H to G [69]. As earwigs collected in CONV orchards were resistant to Op, according to a previous study [20], we focused on the sequences of the emblematic esterases involved in OP resistance, E4 and FE4 esterases. Indeed, in the aphid *Myzus persicae,* it was shown that the level of resistance was correlated with the level of gene amplification. In the most resistant clones, up to 80 times more genes were found for the E4 and FE4 esterases [70,71]. Interestingly, only the level of expression of the CCE E was modulated (between 2.8 to 5-fold) according to the orchard’s origin (Figure 2). CCE clade E included E4 and FE4 esterases of *M. persicae* as well as esterase from the brown planthopper, *Nilaparvata lugens* which was amplified three–seven-fold in OP-resistant strains compared to susceptible strains [69,72]. The amplification of the CCE clade E observed here supports the increase in the constitutive CCE activity measured in earwigs collected in conventional orchards [20], and is consistent with insecticide resistance.

The last detoxification gene that was considered was a glutathione S-transferase from the delta class. GST D was over-expressed for earwigs collected in the orchards with management strategies compared to those collected in the untreated orchards (Figure 2). This is consistent with a higher GST activity measured in *F. auricularia* collected in treated orchards [20]. The highest expression was obtained with IPM management, around six-fold compared to NT. The IPM orchard was treated principally with neonicotinoids, pyrethroids and OPs (Table 2). Delta GSTs have been shown to confer resistance to all these chemical classes in several insect species. In the Asian citrus psyllid, *Diaphorina citri*, the RNAi-mediated silencing of GSTd1 increased the susceptibility to pyrethroids and neonicotinoids [73]. In *Cydia pomonella*, GSTd3 has been demonstrated to confer resistance to lambda-cyhalothrin through passive binding and sequestration [74]. GSTs are also well known for their protecting role against oxidative stress. This raises the question of whether the GST identified here metabolises insecticides or whether it has a role in the general stress response and is therefore induced by any management strategy. Further studies are needed to determine the function of this GST.

## 4. Conclusions

In conclusion, this study is the first molecular investigation on pesticide resistance-associated genes in natural populations of the earwig *F. Auricularia*. We identified here the major pesticide targets and detoxification genes. Whether the point-mutations or over-expression of particular transcripts that we identified are associated with resistance now needs to be ascertained. Future studies therefore face the challenge of improving the laboratory maintenance and production of earwigs to facilitate extensive pesticide-resistance assays on earwigs harbouring specific mutations/over-expressions. Such studies should be extended to a higher number of commercial orchards to fully understand the relationships between insecticide use and resistant strains in *F. auricularia*, as our study was limited to a single sampling by management conditions, and to evaluate their relevance as biocontrol agents. The results could allow us to define how agricultural strategies influence natural enemies’ life-history traits and their potential inter-specific competition. 

## Figures and Tables

**Figure 1 insects-14-00944-f001:**
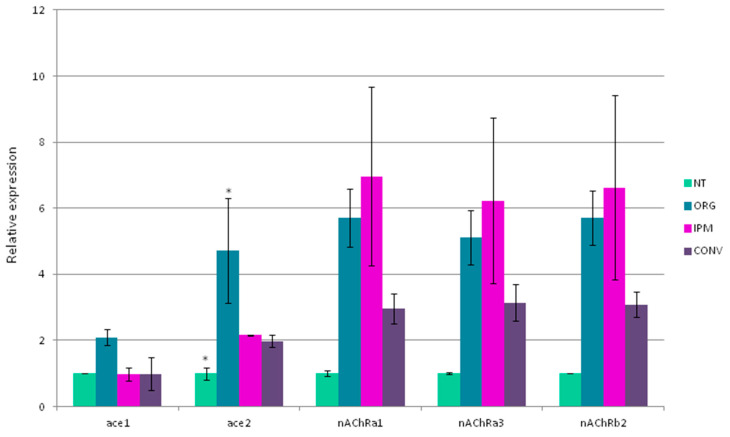
Level of expression of insecticide targets. Gene expression was normalized using the expression of three reference genes (actin, EF1 and GAPDH) and shown as fold-change relative to the expression of earwigs collected in the untreated orchard. ace1; ace2: acetylcholinesterase target of organophosphates and carbamates; nAChRα1; nAChRα3, nAChRβ2: nicotinic acetylcholine receptor (three subunits) target of neonicotinoids and spinosad. Data are mean values of three biological replicates ± SEM, and comparison between samples were performed using One-way ANOVA followed by Tukey test (* *p* < 0.05) indicating significant difference between earwigs from non-treated orchard versus treated orchards.

**Figure 2 insects-14-00944-f002:**
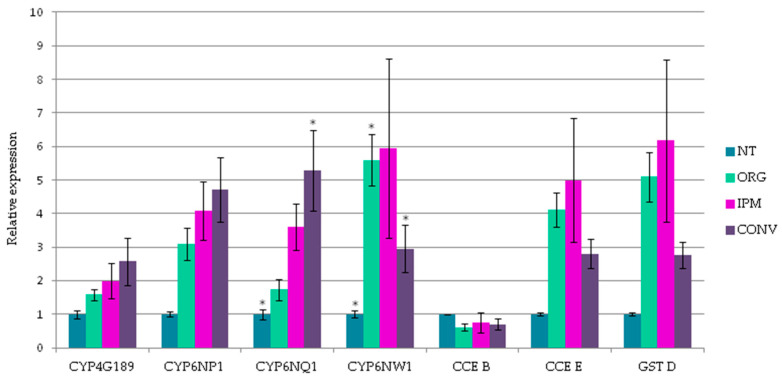
Level of expression of detoxification genes. Gene expression was normalized using the expression of three reference genes (actin, EF1 and GAPDH) and shown as fold-change relative to the expression of earwigs collected in no treated orchard. CYP (cytochromes P450); CCE (carboxylesterases); GST (glutathione-S-transferases). Data are mean values of three biological replicates ± SEM, comparison between samples was performed using one-way ANOVA followed by Tukey test (* *p* < 0.05), indicating significant difference between earwigs from non-treated orchards versus treated orchards.

**Table 1 insects-14-00944-t001:** Genes of interest with their respective GenBank accession number.

Gene Category	Gene Name	GenBank Accession Number
Insecticide target genes	Acetylcholinesterases (Ace)	Ace 1 (MK756004)
		Ace 2 (MK756005)
	Glutamate-gated chloride channel (GluCl)	GluCl (MN942030)
	Nicotinic acetylcholine receptors (nAChR)	nAChR α1(MK756006)
		nAChR α3 (MK756010)
		nAChR β2 (MK756007)
	Ryanodine receptor (RyR)	RyR (MN942031)
	Voltage-sensitive sodium channel (Para)	Para (MN942033)
Detoxification genes	Carboxyl/choline esterase clade B	CCE B ^1^
	Carboxyl/choline esterase clade E	CCE E (MN942032)
	Cytochrome P450, CYP4G189	CYP4G189 (MK756000)
	Cytochrome P450, CYP6NP1	CYP6NP1 (MK756003)
	Cytochrome P450, CYP6NQ1	CYP6NQ1 (MK756001)
	Cytochrome P450, CYP6NW1	CYP6NW1 (MK756002)
	Glutathione S-transferase delta class	GST D (MK756011)
Control genes	Actin	Actin (MK756009)
	Elongation factor 1 (EF1)	EF1 (MK756012)
	Glyceraldehyde-3-phosphate dehydrogenase	GAPDH (MK756008)

^1^ Sequence fragment too small to be deposited in GenBank.

**Table 2 insects-14-00944-t002:** Treatments in organic (ORG), integrated pest management (IPM) or conventional (CONV) orchards over the period from 2008 to 2017.

			Treatment Average Per Year over the Last 10 Years (2008–2017)		
Commercial Product	Active Compound	Chemical Family	ORG	IPM	CONV
Agrimec	Abamectin	Avermectins	0	1	1.3
Affirm, proclaim	Emamectin benzoate				
Karaté K, Okapi liquide	Pirimicarb	Carbamates	0	0	0.5
Coragen	Chlorantraniliprole	Diamides	0	0.6	0.7
Teppeki	Flonicamid	Flonicamid	0	0.9	0.8
Supreme	Acetamiprid	Neonicotinoids	0	1.5	2.8
Calypso, Alanto	Thiacloprid				
Finetyl D, Pyrinex ME, Cuzco	Chlopyrifos	Organophosphates	0	1.8	8.4
Finetyl D	Dimethoate				
Decis protech, Pearl protech, Split protech	Deltamethrin	Pyrethroids	0.125	2	1.4
Karaté zéon, pyrinex ME, Karaté K, Okapi liquide	Lambda-cyhalothrin				
Klartan	Tau-fluvanilate				
Pyrevert	Pyrethrins				
Success 4	Spinosad	Spinosyns	3.125	0	0.4
Delegate	Spirotoram				
Confirm	Tebufenozide	Diacylhydrazine	0	0	0.3
Inségar, precision	Fenoxycarb	Fenoxycarb	0	0.7	0.5
Admiral Pro	Pyriproxyfen	Pyriproxyfen	0	0.3	0
Movento	Spirotetramat	Tetronic acid derivatives	0	0.3	0.4
Envidor	Spirodiclofen				
Delfin	*Bacillus thuringiensis*	*Bacillus thuringiensis*	0	0.2	0
Carpovirusine 2000, Madex twin	CpGV-M1	Granuloviruses	7.125	1	0.6
Carpovirusine Evo2	CpGV-R5				
Neemazal	Azadirachtin A	Limonoid	0.25	0	0
Euphytane 66/gold, oliocin, arb’hiver, seppic ts, ovipron plus, genera, alkakill	Paraffin oil	Mineral oil	1.5	1.4	1.3
Ginko, checkmate CM-XL1000	E-E 8,10dodecadiene-1-ol	Sexual pheromone	0	0.2	0.2
			12.125	11.9	19.6

**Table 3 insects-14-00944-t003:** Overview of mutations detected in the sequence of genes coding for insecticide targets in *F. auricularia*.

Target Gene Name (Insecticide)	Cloned Sequence	Detected Mutation	Corresponding Region
Acetylcholinesterases (organophosphates, carbamates)	Ace1	None	-
	Ace2	Q337K	Near mutation conferring resistance in other insects
Nicotinic acetylcholine receptors (neonicotinoids, spinosad)	α1	P145F	Near binding domain for acetylcholine and neonicotinoids
		E546K	Unidentified
	β2	S6P	Near insecticide binding domain
		G102R	Well-conserved amino acid in proximity to loop A
Voltage sensitive sodium channel (pyrethroids)	Fragment 1 (N-term and domain I)	None	Insertion of 13 nucleotides in the N-term loop
	Fragment 2 (domain II)	None	-
	Fragment 3 (domain III)	A1375V	Not identified yet to confer resistance
		R1879P	
		I1903M	
		E1976K	
Glutamate-gated chloride channel (avermerctins)	GluCl	None	-
Ryanodine receptor (anthranilic diamides)	RyR partial	None	-

## Data Availability

Gene sequences were deposited in GenBank, accession numbers are specified in the text. Raw data are available from authors on request.

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
