# Peer review of "The Molecular Resistance Mechanisms of European Earwigs from Apple Orchards Subjected to Different Management Strategies"

_insects, 2023, doi:10.3390/insects14120944_

Round 1

Reviewer 1 Report (Previous Reviewer 1)

Comments and Suggestions for Authors

The paper by Fricaux et al screens earwigs for mutations and detoxification mechanisms that result in resistance to several classes of insecticide used in apple orchards. The paper is interesting in that it indicates how natural enemies can develop resistance to insecticides in the same way as pests, although earwigs are considered major pests of stone fruits in some regions. The authors have been rigorous in the molecular aspects of their study, but less so when it comes to sampling different types of orchard. This needs to be made very clear in the manuscript - for example, the ANOVAs are based on N = 1 for each type of orchard - therefore, nothing can really be said about resistance in the context of orchard management - these aspects should be toned down or the shortcomings further high-lighted.

Comments on the Quality of English Language

The manuscript is full of awkward sentences and requires support from a native English speaker to improve readability.

Author Response

Reviewer 2 Report (Previous Reviewer 2)

Comments and Suggestions for Authors

I accept the corrections and improvement the Authors have done to the manuscript. This time, after the Authors' effort to follow my suggestions, my recommendation is: accept in present form.

Author Response

Reviewer 3 Report (Previous Reviewer 3)

Comments and Suggestions for Authors

I have two questions about this manuscript:

1. the authors showed specific mutation in the emzymes which have important roles. Could the authors show the enzymes structure, such as different domains?

2. how the authors design the primers for qRT-PCR? 

Author Response

Reviewer 4 Report (New Reviewer)

Comments and Suggestions for Authors

Fricaux et al. research on the various resistance factors to the pesticides of the earwigs, one of the beneficial insects, to expect the future use in the orchards. Since this manuscript is well designed and fits with the scope of the journal, attracting the readers. I have one suggestion and find one typo.

line 199-213 I think the section3.1 "Comparing treatments between orchards" and Table 2 should be included in Materials and Methods, because these data are only the conditions of the experiments and are not obtained from the experiments.

L416 piperonil -^> piperonyl 

Comments on the Quality of English Language

well written

Author Response

Response to reviewers

We would like to thank the reviewers for their valuable and critical comments which helped us to improve our manuscript. We hope these revisions meet your approval.

Please find below point by point responses to the reviewers’ comments.

Reviewer 4:

Fricaux et al. research on the various resistance factors to the pesticides of the earwigs, one of the beneficial insects, to expect the future use in the orchards. Since this manuscript is well designed and fits with the scope of the journal, attracting the readers. I have one suggestion and find one typo.

We thank the reviewer for valuable comment. 

line 199-213 I think the section3.1 "Comparing treatments between orchards" and Table 2 should be included in Materials and Methods, because these data are only the conditions of the experiments and are not obtained from the experiments.

We agree that these are not experimental results, but we would like to keep this description in the results and discussion section because we are discussing the differences between the orchards. We feel that this step is important for understanding the results that follow and some readers do not read the materials and methods section.

L416 piperonil -^> piperonyl 

Correction was done.

Round 2

Reviewer 1 Report (Previous Reviewer 1)

Comments and Suggestions for Authors

It's a little disappointing to see such minor changes to the manuscript when a major overhaul was suggested. The authors have indicated in a number of places that they require more orchards to be sampled to be able to infer about resistance mechanisms and practices; however, they have not removed all such inferences (for example lines 15-16!). The study has some excellent qualities and excellent outputs/advances, particularly as regards the diversity of mechanisms and the possibilities to detect these - but the paper will probably be overlooked by pest managers and certainly overlooked by any competent ecologists as a non-replicated study that is structured around seeking a link between practices and the genetics of resistance. This goes further if we think about earwigs collected under cardboard and the possibilities that these may be genetically related - perhaps even siblings. 

Comments on the Quality of English Language

Some awkward sentences - will probably be fixed at editing

Author Response

We would like to thank the reviewers for their valuable and critical comments which helped us to improve our manuscript. We hope these revisions meet your approval.

Please find below point by point responses to the reviewers’ comments.

Reviewer 1:

It's a little disappointing to see such minor changes to the manuscript when a major overhaul was suggested. The authors have indicated in a number of places that they require more orchards to be sampled to be able to infer about resistance mechanisms and practices; however, they have not removed all such inferences (for example lines 15-16!). The study has some excellent qualities and excellent outputs/advances, particularly as regards the diversity of mechanisms and the possibilities to detect these - but the paper will probably be overlooked by pest managers and certainly overlooked by any competent ecologists as a non-replicated study that is structured around seeking a link between practices and the genetics of resistance. This goes further if we think about earwigs collected under cardboard and the possibilities that these may be genetically related - perhaps even siblings. 

We are aware of the limitations of our study. We do not have the means to carry out a new sampling on a larger number of orchards. We have modified lines 15 and 16 as requested by reviewer 1. We have indicated the limitations several times in the Results and Discussion section and in the conclusion:

  • Line 361-362 “but this needs to be confirmed by sampling in large number of orchards”.
  • Line 375-376 “even if a larger number of orchards representative of each management condition would be desirable”.
  • Line 461-464 “Such studies should be extended to higher number of commercial orchards to fully understand relationships between insecticide use and resistant strain in auricularia, as our study was limited to a single sampling by management condition ».

This manuscript is a resubmission of an earlier submission. The following is a list of the peer review reports and author responses from that submission.

Round 1

Reviewer 1 Report

Comments and Suggestions for Authors

The study screened earwigs from four apple orchards in France for potential mechanisms of resistance to a series of insecticides. The sites included an organic, an IPM and a conventional orchard. The authors documented insecticide use at the sites (based on questionnaires) and sampled earwigs. The earwigs were then screened for the presence of resistance mutations, for expression of genes of interest, and for the expression of detoxification enzymes. The authors report a series of resistance-related indicators and compare across orchard treatments.

The paper presents some new information, and does suggest that there are a diversity of resistance mechanisms among the natural enemies of orchard pests; however, the authors have focused almost entirely on the molecular component of the research and then attempted to expound to an ecological level. I suggest that the authors should remain within the bounds of their study as dictated by their choice of methods. For example, earwigs were collected from a single orchard of each type (according to the methods) - which is insufficient to allow a statistical comparison of potential treatment effects (the study is pseudoreplicated). The authors also used very few individuals for their analyses - which gives very little indication of the variability in measured effects over time or spatially. This is only a problem if the authors wish to relate their results to management - which is what they have done in their manuscript. I would suggest that any reference to management effects must be reduced.

Furthermore, the authors offer no indication of whether the insects were actually resistant to any of the proposed chemicals. This is another major short-coming. There does seem to be an accompanying paper related to chlorpyrifos, but this needs further information in this present paper.

Overall, these two major short-comings substantially weaken the paper and I would suggest that the authors return to their results and rethink about the value of these results and the story that they can deliver based on their sampling regime. This could simply be an indication of the diversity of resistance mechanisms from earwig populations - without relation to management.

The management issue becomes more worrisome when the authors propose that resistant earwigs might be useful for integrated pest management; however, the authors should remember that earwigs are omnivorous insects that are often the main pests in stone fruits, and several studies have shown that pesticide resistance of non-target organisms can lead to insecticide overuse - so liberating resistant populations is probably not a good idea.

Some minor points:

For clarity, it would be better to separate the results and discussion - it is difficult to keep tract of the actual results due to the volume of information presented.

Figure 1 is not required – also, it should be acknowledged/referenced if not original;

369 – earwigs from orchards exposed to substantial treatments – maybe they just survive better because of trade-offs in resistance against other insecticides, the mechanism needs more thought – also contradicts some studies as suggested in 345-347;

371 – I don’t see the connection – explain the link between your results and those from ref 49 in the context of these sentences

377 – orchard sampling was NOT appropriate because of a lack of statistical power (and see my comments above) – i.e., obvious trends but not significantly different

Conclusions not in line with abstract - but better than abstract

Comments on the Quality of English Language

The paper is well-written in terms of English language; some of the references need to be better explained in the context of the results or discussion.

Reviewer 2 Report

Comments and Suggestions for Authors

Thank you for the opportunity to review the manuscript. I find it very interesting, justified, valuable and corresponding to the real ecotoxicological needs and leading topics of this branch of science. I think it is worth publishing, however after major revision. Unfortunately there are some significant weak points at this stage. Please find my detail comments below:

-          Line 21 (Abstract): “To date, apples are among the most treated with up to 35 chemical treatments per year” – what scale the Authors mean: global? European? France?

-          Line 102: delete ‘a’

-          Lines 112-123: Aims of the study should be clearly defined. They are too general. In fact, the Authors  are not indicated them at all. The Authors rather describe their action than indicate questions/scientific problems. That is why the sentences in concluding section (see below) do not sound as conclusions but as a summary. The study is too interesting to leave it so illegible, indistinct. That is why I more than suggest to profoundly rethink the two paragraph so they correspond to each other.

-          Lines 193-194: The Authors should present the sources of the data of the use of chemicals in the model orchards or the method they used to identify them.

-          Results and discussion: I would strongly suggest a graphical/tabular summary of which genes and how many mutations - it is in the text but gets lost among the descriptions of the results and discussions

-          Caption to Figures 2 and 3: As figures should be self-explaining, therefore the caption should be completed with the following information: insecticide targets should be named exactly. Also, Authors should explain what kind of differences, are indicated by the asterisks, and what kind of differences are indicated by letters

-          Lines 452-463: Conclusions. As mentioned above, the conclusions should clearly refer to aims/hypotheses. The Authors write: “We hypothesized that the repeated and diverse treatment used in orchards may result in the selection of various pesticide resistance traits in earwigs”. The authors do not refer to it at all in this section. This is the main weak point of the paper because the readers should be able to find whether the hypothesis was verified/falsified.

Reviewer 3 Report

Comments and Suggestions for Authors

The concept behind the experiment is straightforward. The reader without expertise could easily understand. The results the authors showed could help other researchers to better understand the molecular resistance mechanism of  earwigs. There are some comments about this manuscript. 

1. Some sentences need to add citation.

2. The figure 2 and 3 need to be improved.

Comments on the Quality of English Language

The quality of English language is OK.